# Validation of an Artificial Intelligence-Based Model for Early Childhood Caries Detection in Dental Photographs

**DOI:** 10.3390/jcm13175215

**Published:** 2024-09-03

**Authors:** Julia Schwarzmaier, Elisabeth Frenkel, Julia Neumayr, Nour Ammar, Andreas Kessler, Falk Schwendicke, Jan Kühnisch, Helena Dujic

**Affiliations:** 1Department of Conservative Dentistry and Periodontology, University Hospital, Ludwig-Maximilians University of Munich, 80336 Munich, Germany; julia.schwarzmaier@web.de (J.S.); lisafrenkel04@gmail.com (E.F.); julia.neumayr97@web.de (J.N.); nour.ammar@outlook.de (N.A.); andreas.kessler@uniklinik-freiburg.de (A.K.); falk.schwendicke@med.uni-muenchen.de (F.S.); h.dujic@med.uni-muenchen.de (H.D.); 2Department of Prosthetic Dentistry, Center for Dental Medicine, University Hospital Freiburg, 79106 Freiburg, Germany

**Keywords:** dental caries, early childhood caries, artificial intelligence, diagnosis

## Abstract

**Background/Objectives**: Early childhood caries (ECC) is a widespread and severe oral health problem that potentially affects the general health of children. Visual–tactile examination remains the diagnostic method of choice to diagnose ECC, although visual examination could be automated by artificial intelligence (AI) tools in the future. The aim of this study was the external validation of a recently published and freely accessible AI-based model for detecting ECC and classifying carious lesions in dental photographs. **Methods**: A total of 143 anonymised photographs of anterior deciduous teeth (ECC = 107, controls = 36) were visually evaluated by the dental study group (reference test) and analysed using the AI-based model (test method). Diagnostic performance was determined statistically. **Results**: ECC detection accuracy was 97.2%. Diagnostic performance varied between carious lesion classes (noncavitated lesions, greyish translucency/microcavity, cavitation, destructed tooth), with accuracies ranging from 88.9% to 98.1%, sensitivities ranging from 68.8% to 98.5% and specificities ranging from 86.1% to 99.4%. The area under the curve ranged from 0.834 to 0.964. **Conclusions**: The performance of the AI-based model is similar to that reported for the internal dataset used by developers. Further studies with independent image samples are required to comprehensively gauge the performance of the model.

## 1. Introduction

Early childhood caries (ECC) is among the most prevalent diseases worldwide [1,2,3,4]. Defined as the presence of one or more noncavitated or cavitated carious lesions, and missing or filled tooth surfaces due to caries in any deciduous tooth in a child under the age of six, ECC is often associated with a loss of dental functionality and aesthetics, pain due to abscesses, underweight, or reduced quality of life if left untreated [5]. Furthermore, the risk of caries development in the permanent dentition is increased [6]. For experienced professionals, the diagnosis of ECC is usually an immediate visual diagnosis. The diagnostic assessment is made using various classification systems, which consider either the extent of affected deciduous teeth in relation to age [4,7] or the distribution pattern of carious lesions in the dentition [8]. The established visual criteria facilitate precise diagnostic assessment of all teeth or tooth surfaces [9,10,11,12,13].

As part of ongoing efforts to digitise and automate caries diagnostics, several studies have reported on the use of deep learning models for caries detection on intraoral photographs [14,15,16,17,18] and dental radiographs [19,20,21]. Although these studies differ in methodology, model type, and dataset size, the reported results show promising diagnostic performance [14,15,18,19,21]. Specifically in paediatric dentistry, AI-based models have been explored not only for caries detection on intraoral photographs [17] but also for the identification and numbering of deciduous and permanent teeth [22,23] as well as the identification of mesiodens on radiographs [24,25,26].

Despite these advances, most studies report only internal validation data—results obtained during the initial development and validation using the test dataset—and the models developed are often not freely accessible [15,16,17,18]. Furthermore, the lack of external validation limits their applicability to different clinical settings. In contrast to this, Felsch et al. [14] published a model that uses artificial intelligence (AI) to automatically detect, classify, localise and segment caries on photographs of teeth. This AI-based model is freely accessible and can currently be used for image analysis without restriction. This accessibility is crucial as it allows for external validation using independent imaging data that were not part of the model’s training. Thus, the model’s diagnostic performance and, by extension, its generalisability can be tested on new, unseen data, enabling a comparison with the internal validation results. The aim of the present study was to validate the AI-based model [14] using independent image data of anterior deciduous teeth. It was hypothesised that there would be no significant differences between the internal and external validation accuracies.

## 2. Materials and Methods

This investigation was conducted following the recommendations of the STARD Steering Committee (Standards for Reporting of Diagnostic Accuracy Studies) and the recently published recommendations for the design and conduct of studies using AI methods in dental research [27,28].

### 2.1. Dental Images

A total of 143 anonymised dental photographs of anterior deciduous teeth were selected from an established external image database created for clinical documentation and training purposes, which includes a wide range of clinical presentations of ECC. The images were captured by an experienced dentist (JK) using standard procedures. Specifically, each intraoral image was taken with a professional single-reflex lens camera (D300, D7100, or D7200 with a Nikon Micro 105 mm lens, Nikon, Minato (Tokio), Japan) equipped with a macro lens and a macro flash (EM-DG 140, Sigma, Kawasaki (Kanagawa), Japan). Before taking the photographs, the teeth were cleaned and dried. All photographs had the following parameters: aspect ratio of 3:2, minimum resolution of 2784 × 1856 pixels without compression, jpeg format and RGB colour space. None of the photographs were generated using AI methods, edited or manipulated. The selection was conducted independently of all other analyses by an examiner with over 20 years of clinical experience, including extensive practice in paediatric dentistry. The process followed established clinical criteria for detecting and diagnosing ECC, utilizing classification systems that assess the distribution pattern of lesions in the dentition [8]. These criteria ensured that the selected images accurately and comprehensively reflected the relevant clinical manifestations.

### 2.2. Dental Image Evaluation (Reference Test)

The selected photographs (*n* = 143) were evaluated in detail by the dental study group of five dentists (JS, EF, JN, HD, JK). The dataset included images from different stages of ECC (*n* = 107) and caries-free dentitions (*n* = 36). In addition to the basic decision as to whether ECC was present, all existing caries findings were carefully evaluated. In detail, each carious lesion was classified according to the ICDAS/UniViSS criteria: caries-related noncavitated opacities in the enamel, caries-related breakdown of the chalky or brown-stained enamel surface in the form of a microcavity, greyish translucency indicating underlying dentin involvement, cavitation with visible dentin involvement, and extensive cavities up to the complete destruction of the tooth crown [12,29,30]. In case of differing opinions, the finding was discussed within the dental study group until a consensus decision was reached among the five dentists. The documented findings served as the reference test.

Prior to this study, the investigators had completed a one-day theoretical training session on the assessment of ECC and noncavitated and cavitated carious lesions under the supervision of an experienced principal investigator. The training provided information on the study design, indices and diagnostic principles.

### 2.3. AI-Based Image Evaluation (Test Method)

For automated image evaluation, the freely accessible web tool (https://demo.dental-ai.de/, accessed on 21 August 2024), which can detect, classify, localise and segment carious lesions on dental photographs, was used for AI-based analysis [14]. The carious lesion classification was based on the ICDAS and UniViSS criteria [12,29,30]. Specifically, the web tool distinguishes between the following caries scores: 1—noncavitated carious lesion, 2—greyish translucency/microcavity, 3—caries-related cavitation and 4—destructed tooth (Figure 1). All selected anonymised images (*n* = 143) were uploaded individually to the aforementioned website. Once uploaded, the images were cropped to optimally display the four maxillary incisors (area of interest, Figure 1). Automated image analysis was then performed, and all potential classes of findings were marked as coloured pixels. Each colour corresponds to a specific type of finding identified by the AI-based model. Multiple adjacent pixels of the same colour formed a pixel cloud or segment representing a localised area of interest. For example, green pixels indicate cavitated lesions, while blue pixels represent greyish translucencies (Figure 1). For dental images showing caries-free dentition, no segment was highlighted by the AI-based model (Figure 1). Each AI-based image analysis result was captured and saved as a screenshot for later independent dental evaluation (Figure 1). After two weeks, the findings and segments marked by the AI-based model were assessed separately by the dental study group. A total of 261 segments were identified on all included dental photographs as AI outputs. In addition, localisation and segmentation of all caries segments from the AI-based image analysis were assessed for their correctness. Caries segment localisation could be either correct or incorrect. Here, at least one pixel had to be located in the carious lesion. When assessing the marked segments (AI outputs), a distinction was made between incorrect, partially correct and fully correct segments. If >90% of the actual caries extent was recognised by the AI, this was classified as fully correct. If the AI recognised most of the caries extent (<90%), this corresponded to the partially correct segmentation and for caries segments outside the actual lesion, the segmentation was classified as incorrect. The segments were assessed as estimates, as exact values could not be determined.

### 2.4. Data Management and Statistics

For this project, an entry form was used to document all diagnostic findings directly (EpiData Manager and EpiData Entry Client, V4.6.0.6, EpiData Association, Odense, Denmark, http://www.epidata.dk, accessed on 21 August 2024). The dataset was exported to an Excel spreadsheet (Excel 2019, Microsoft Corporation, Redmond, WA, USA) following data collection and prepared for statistical exploration. The diagnostic performance of the test method in comparison to the reference test was calculated using Python V3.8.5 (http://www.python.org, accessed on 21 August 2024). The true-positive (*TP*), true-negative (*TN*), false-positive (*FP*) and false-negative (*FN*) rates were calculated as key figures from contingency tables [31]. Based on this, accuracy (*ACC*), sensitivity (*SE*), specificity (*SP*), and positive and negative predictive values (*PPV*, *NPV*) were determined:(1)ACC=(TP+TN)/(TP+TN+FP+FN)
(2)SE=TP/(TP+FN)
(3)SP=TN/(TN+FP)
(4)PPV=TP/(TP+FP)
(5)NPV=TN/(TN+FN)

In addition, the area under the receiver operating characteristic (ROC) curve (*AUC*) was determined.

## 3. Results

The AI-based diagnostic model correctly diagnosed caries in terms of ECC on the photographs in a total of 104 images and thus achieved a *SE* of 97.2% (Table 1). Out of 36 photographs with caries-free deciduous teeth, 35 were correctly recognised, yielding a *SP* of 97.2%. This resulted in an overall diagnostic accuracy of 97.2% (Table 1).

The chosen dental photographs included multiple carious lesions which, therefore, required further analyses. The cross-tabulation of all caries diagnostic decisions from the AI-based evaluation (test method) and the visual examination (reference standard) can be taken from Table 2. The *TP*, *TN*, *FP* and *FN* rates are summarized for each caries class in Table 3. The diagnostic performance parameters of the external validation of the AI-based model are listed in Table 4. In detail, the *ACC* for the test method ranged from 88.9% (cavitation) to 98.1% (destructed tooth). The *SE* values ranged from 68.8% (greyish translucency/microcavity) to 98.5% (noncavitated carious lesion). The *SP* values ranged from 86.1% (noncavitated carious lesion) to 99.4% (cavitation).

Figure 2 shows the ROC curves for the chosen caries classes and images with a caries-free dentition. The corresponding *AUC* values, which ranged from 0.834 for greyish translucency/microcavity to 0.964 for healthy dentition (Table 4), provide a summary measure of the model’s overall diagnostic performance for each caries class.

The results for caries lesion localisation and segmentation are shown in Table 5 and Table 6. A total of 143 images were analysed with the AI-based model, whereby this sample showed 226 carious lesions (86.6%) and 35 caries-free images (13.4%). In the case of existing caries, 220 segments (84.3%) were correctly localised; only 6 segments (2.3%) were incorrectly localised (Table 5). The AI-based model correctly predicted segmentation in 114 cases (43.6%). The segmentation of 101 cases (38.7%) was partially correct and in 11 cases (4.3%) the prediction was outside the existing carious lesion (Table 6).

## 4. Discussion

The study showed that the freely accessible AI-based model can detect carious lesions on dental photographs from photographs with ECC with a high diagnostic accuracy of 97.2% (Table 1). The documented *AUC* values from the ROC curves for the classification of the included caries categories ranged from 0.834 to 0.964 (Figure 2), indicating an encouraging result for automated caries detection on photographs of anterior deciduous teeth. Furthermore, the AI-based model demonstrated high sensitivity and specificity for detecting noncavitated lesions (*SE* 98.5%, *SP* 86.1%) and destructed teeth (*SE* 92.9%, *SP* 99.1%), showing a promising performance in identifying both early-stage and advanced carious lesions. However, the model’s lower sensitivity for cavitated lesions (*SE* 71.7%) and greyish translucency/microcavity (*SE* 68.8%) indicates that despite high specificity, a number of cases in these categories might be overlooked. A comparison of the documented data on external validity (Table 4) with the published results on internal validity [14] showed very good agreement based on the *ACC* values. The *ACC* values from the internal [14] and external validation (Table 4, Figure 2) datasets were 90.1% and 89.3% (noncavitated carious lesion), 99.0% and 96.2% (greyish translucency/microcavity), 95.9% and 88.9% (cavitation), and 99.0% and 98.1% (destructed tooth), respectively. Based on these findings, the initial hypothesis that there would be no difference between the external and internal accuracies can be confirmed.

Moreover, when considering other diagnostic studies on automated caries detection and classification in dental photographs, a more inconsistent picture emerges. In a study project that included smartphone images of ECC patients, the model performance was assessed but not supported by detailed validation data on caries detection and classification [32]. In another paper, Zhang et al. [33] published data on the internal validity of their approach to AI-based caries detection on dental photographs. However, the internal validation data indicated that the model performance appears to be substantially lower than that of the model presented by Felsch et al. [14]. Specifically, the model by Zhang et al. [33] achieved an imagewise *SE* of 81.9% and a boxwise *SE* of 64.6% [33]. In another project [16], four different AI-based models for caries detection were developed, and their internal validity was assessed. The *ACC* values varied depending on the deep learning model, ranging from 60.7% to 68.8% for noncavitated lesions and from 81.0% to 87.4% for cavitated lesions. Similarly, Park et al. [34] reported internal validity data of a similar magnitude for caries detection, with *ACC* values of 75.8% without object extraction and 81.3% with object extraction. In addition, further projects on caries detection and classification on dental photographs were published, which had fewer methodological similarities with the aforementioned studies. Moharrami et al. [35] published an overview of the currently available projects and noted that automatic dental caries detection using AI may provide objective verification of clinicians’ diagnoses. However, future studies should use more robust designs, standardized metrics, and focus on caries detection and classification metrics.

A special feature of the freely accessible AI-based model [14] is that, in addition to caries detection and classification, the evaluated images include information on lesion localisation and segmentation. This is accompanied by the simultaneous visualisation of different caries classes in one image as part of a carious lesion (Figure 1). Compared to conventional clinical examination, this is an interesting feature of AI-based image evaluation that has not yet been considered. Evaluating the information on lesion localisation (Table 5), 220 out of a total of 226 cases were correct and thus in the range of high diagnostic performance. Only a few cases (*n* = 6) showed incorrect localisation. This situation was somewhat less favourable for the segmentation performance of the AI-based model, as only 114 of 226 lesion segments were fully correct (Table 6). The model performed best in the segmentation of noncavitated carious lesions. Fully destructed teeth and dentin cavitation were proportionally less likely to be segmented fully correctly; i.e., the displayed areas deviated from the actual extent of caries in the dentist’s assessment. However, it should be noted that such evaluations are based on analysing each individual image at the pixel level, offering the dentist an unprecedented level of precision. Further improvements, which could be achieved by continuously fine-tuning the AI-based model, are desirable at this point.

Finally, the strengths and limitations of this study project should be discussed. This study represents an external validation of a recently published and freely available AI-based method for automated image analysis. For this purpose, this study utilised image data that had not been used in the development of the model. This allowed for an independent verification of the AI-based model. The images used in this study were professionally captured and of comparable quality to the training dataset of the AI-based model [14]. On the one hand, it can be assumed that high quality has a positive effect on AI-based image evaluation; on the other hand, it can be argued that high-quality images may not be available in all clinical situations. This ultimately raises the question to what extent a correct diagnosis can be made on low-quality images. However, it should also be noted that the object of interest needs to be correctly depicted [36], as otherwise, a diagnostic evaluation is potentially impossible. Provided the images are of good quality, a valid diagnostic assessment is ultimately possible [36,37]. Furthermore, the photographs only showed anterior deciduous teeth in this study. Therefore, no conclusions can be drawn about the diagnostic performance in the posterior region, for which the AI-based model was also developed [14]. Moreover, the fact that the included images were anonymized and lacked clinical data can be taken into account as a limitation. This is relevant with regard to the potential impact of remineralisation on the visual characteristics of the lesions, as remineralised areas may appear less prominent or have a different structure compared to active lesions. It remains unclear to what extent this process could affect the diagnostic performance of the AI-based model. Given the number of included and available photographs both with and without ECC for this study, it should be emphasized that although a larger number of images would be desirable, obtaining such a dataset with high-quality images seems rather challenging. Nevertheless, these points should be considered in future studies to verify the reliability and generalizability of the tested AI-based model. Another desirable feature that is not yet possible with the validated AI-based model is the analysis of images with regard to a more precise ECC type [4,7,8]. The currently available AI-based model [14] enables only the detection, classification, localisation and segmentation of caries and enamel hypomineralisations. Furthermore, it is currently unable to analyse direct or indirect restorations, dental trauma and other findings, such as plaque and discolouration. Additionally, the evaluation of dental photographs with multiple teeth is not yet fully functional, requiring the area of interest to be centred using the crop tool for valid image analysis. Ultimately, this approach was also part of the methodology of this diagnostic study, which relied on visual consensus diagnoses as the reference test, without the inclusion of any histological investigation techniques to provide details about caries characteristics. Lastly, while detailed evaluations of carious lesions cannot be performed independently of the AI-based diagnostic output, the study group aimed to minimise potential evaluation bias by making consensus decisions on all diagnoses.

## 5. Conclusions

In comparison to the initially published internal validation data [14], we found similar results in our external validation of dental photographs of cases with and without ECC. This underlines the diagnostic quality of the AI-based model presented by Felsch et al. [14] for caries detection, classification, localisation and segmentation. However, additional studies with independent image samples are needed to comprehensively describe the performance of the model.

## Figures and Tables

**Figure 1 jcm-13-05215-f001:**
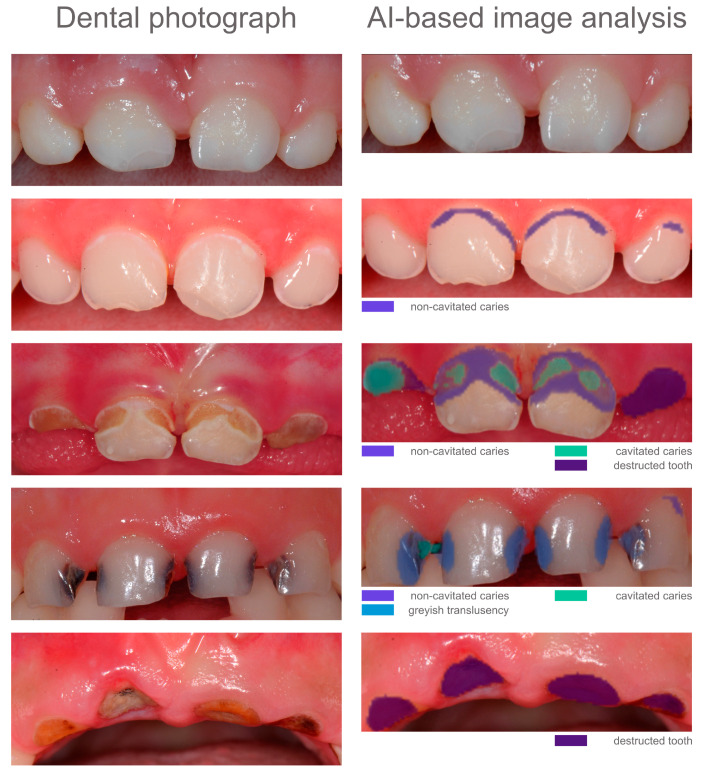
Exemplary photographs (**left**) and corresponding AI-based image analysis (**right**).

**Figure 2 jcm-13-05215-f002:**
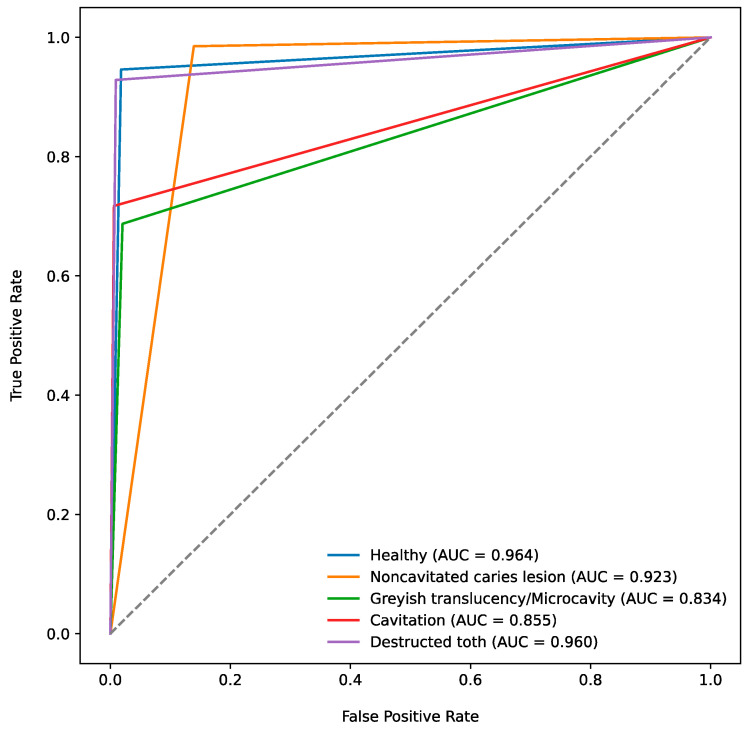
Receiver operating characteristic (ROC) curves and corresponding area under ROC curve (*AUC*) values for the test method and the chosen caries classes. Curves above the dashed line indicate that the proportion of correctly classified images is greater that the proportion of incorrectly classified ones.

**Table 1 jcm-13-05215-t001:** Cross-tabulation including diagnostic performance parameters for the image-related AI-based evaluation (test method) presented in rows in relation to the visual consensus diagnosis by the dental workgroup (reference test) presented in columns. In this analysis, the diagnostic performance for caries detection per image (*n* = 143) was considered.

Caries Detection		Visual Evaluation (Reference Test)		
		Healthy	Caries	
AI-based evaluation (Test method)	Healthy	35	3	*NPV* = 92.1%
	Caries	1	104	*PPV* = 99.0%
		*SP* = 97.2%	*SE* = 97.2%	*ACC* = 97.2%

**Table 2 jcm-13-05215-t002:** Contingency table of the AI-based image evaluation data (test method) presented in rows and visual consensus diagnosis data (reference test) presented in columns for all caries segments. This tabulation includes all diagnoses (*n* = 261) from all images (*n* = 143), with multiple findings per image possible.

Caries Classification		Visual Evaluation (Reference Test)					
		Healthy *	Noncavitated Caries Lesion	Greyish Translucency/ Microcavity	Cavitation	Destructed Tooth	∑
AI-based evaluation (Test method)	Healthy *	35	1	0	2	1	39
	Noncavitated caries lesion	2	66	5	19	1	93
	Greyish translucency/Microcavity	0	0	11	5	0	16
	Cavitation	0	0	0	71	1	72
	Destructed tooth	0	0	0	2	39	41
	∑	37	67	16	99	42	261

* Caries-free surfaces are indicated by the absence of segment markings.

**Table 3 jcm-13-05215-t003:** The table illustrates the true-positive (*TP*), true-negative (*TN*), false-positive (*FP*) and false-negative (*FN*) rates in relation to the used caries classes. The tabulation summarizes all diagnoses (*n* = 261) from all photographs (*n* = 143) as shown in the contingency table.

	Healthy	Noncavitated Caries Lesion	Greyish Translucency/ Microcavity	Cavitation	Destructed Tooth
	*n* (%)	*n* (%)	*n* (%)	*n* (%)	*n* (%)
True positives	35 (13.4)	66 (25.3)	11 (4.2)	71 (27.2)	39 (14.9)
True negatives	220 (84.3)	167 (64.0)	240 (92.0)	161 (61.7)	217 (83.1)
False positives	4 (1.5)	27 (10.3)	5 (1.9)	1 (0.4)	2 (0.8)
False negatives	2 (0.8)	1 (0.4)	5 (1.9)	28 (10.7)	3 (1.2)
∑	261 (100.0)	261 (100.0)	261 (100.0)	261 (100.0)	261 (100.0)

**Table 4 jcm-13-05215-t004:** Diagnostic performance of the AI-based evaluation for each of the chosen caries classes. The calculations are based on all *TP*, *TN*, *FP* and *FN* rates (*n* = 261 in each caries class) from all photographs (*n* = 143) as shown in Table 3.

	Healthy	Noncavitated Caries Lesion	Greyish Translucency/ Microcavity	Cavitation	Destructed Tooth
*ACC* (in %)	97.7	89.3	96.2	88.9	98.1
*SE* (in %)	94.6	98.5	68.8	71.7	92.9
*SP* (in %)	98.2	86.1	98.0	99.4	99.1
*PPV* (in %)	89.7	71.0	68.8	98.6	95.1
*NPV* (in %)	99.1	99.4	98.0	85.2	98.6
*AUC*	0.964	0.923	0.834	0.855	0.960

Abbreviations: *ACC*, accuracy; *SE*, sensitivity; *SP*, specificity; *PPV*, positive predictive value; *NPV*, negative predictive value; *AUC*, area under the receiver operating characteristic (ROC) curve.

**Table 5 jcm-13-05215-t005:** Results of the evaluation of AI-based caries localisation. The analysis included all diagnoses (*n* = 261) from all images (*n* = 143). * No caries localisation was possible.

	Healthy *	Noncavitated Caries Lesion	Greyish Translucency /Microcavity	Cavitation	Destructed Tooth	∑
	*n* (%)	*n* (%)	*n* (%)	*n* (%)	*n* (%)	*n* (%)
Incorrect	4 (1.5)	2 (0.8)	-	-	-	6 (2.3)
Correct	-	91 (34.9)	16 (6.1)	72 (27.6)	41 (15.7)	220 (84.3)
Healthy *	35 (13.4)	-	-	-	-	35 (13.4)
∑	39 (14.9)	93 (35.7)	16 (6.1)	72 (27.6)	41 (15.7)	261 (100.0)

**Table 6 jcm-13-05215-t006:** Summary of the evaluation of AI-based caries segmentation. The analysis included all diagnoses (*n* = 261) from all images (*n* = 143). * No caries segmentation was possible.

	Healthy *	Noncavitated Caries Lesion	Greyish Translucency /Microcavity	Cavitation	Destructed Tooth	∑
	*n* (%)	*n* (%)	*n* (%)	*n* (%)	*n* (%)	*n* (%)
Incorrect	4 (1.5)	2 (0.8)	1 (0.4)	3 (1.2)	1 (0.4)	11 (4.3)
Partially correct	-	31 (11.9)	9 (3.4)	36 (13.8)	25 (9.6)	101 (38.7)
Fully correct	-	60 (23.0)	6 (2.3)	33 (12.6)	15 (5.7)	114 (43.6)
Healthy *	35 (13.4)	-	-	-	-	35 (13.4)
∑	39 (14.9)	93 (35.7)	16 (6.1)	72 (27.6)	41 (15.7)	261 (100.0)

## Data Availability

The AI-based model is available as a web application and can be accessed at https://dental-ai.de (accessed on 21 August 2024). The data that support the findings of this study are available from the corresponding author upon reasonable request.

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
