# Peer review of "Validation of an Artificial Intelligence-Based Model for Early Childhood Caries Detection in Dental Photographs"

_jcm, 2024, doi:10.3390/jcm13175215_

Round 1

Reviewer 1 Report

Comments and Suggestions for Authors

Title: The current title is too lengthy. A suggested revision is: Validation of an Artificial Intelligence-Based Model for Early Childhood Caries Detection.

Introduction: The introduction needs to be revised to clearly present the current state of research and identify gaps in the literature regarding AI-based models for caries detection. It should also thoroughly discuss the results. Is this the only AI-based model developed for this purpose?

Specific Feedback:

·       Line 50: The hypothesis states there would be no differences between internal and external validation accuracies. Could the authors clarify what is meant by internal and external validation in this context?

·       Line 72: It mentions that all existing caries findings were carefully evaluated. Could you specify who conducted this evaluation?

·       Line 61: It notes that intraoral images were taken with a professional single-lens reflex camera with a macro lens and flash. Please provide details about the camera’s manufacturer.

·       Line 63: The text states that teeth were cleaned and dried as well as possible before taking photographs. Please check the English language for clarity.

·       Lines 70-72: There are 143 selected photographs; could you justify why this specific number was chosen? Additionally, 36 caries-free dentitions are mentioned—were these used as controls? If so, why were exactly 36 chosen? Who constitutes the dental study group mentioned, and who made the consensus decision?

·       Line 90: Potential classes of findings were marked as colored pixels. Please explain how these colored pixels were generated and what the colors represent, particularly the blue color in the fourth image down.

·       Tables: The results tables are confusing and difficult to interpret. It’s unclear which data pertain to AI-based evaluation and which to visual evaluation (reference test). The cross-tabulation of AI-based image evaluation relative to visual consensus diagnosis needs to be presented more clearly.

·       Line 133: The tabulation includes all diagnoses (N=261) from all images. Please explain the source of these 261 diagnoses before presenting them in the table title.

·       Table 3: It is unclear whether the results pertain to AI-based evaluation or visual evaluation (reference test). The tables need clearer explanations.

·       Lines 136-138: Diagnostic performance parameters are summarized in Table 3, with accuracy (ACC) ranging from 88.9% to 98.1% and sensitivity (SE) values ranging from 68.8% to 98.5%. It is not clear how these values were calculated. The data and calculations need to be clarified, and the tables should be re-organized to clearly distinguish between AI-based and visual evaluation data.

·       Figure 2: The figure shows ROC curves and corresponding AUC values. Please explain what this figure demonstrates and what additional information it provides.

Discussion:

·       The discussion needs to be rewritten for clarity and grammatical accuracy.

·       Line 166: The term "internal validity" is mentioned in relation to published results. More detail is needed on what "internal validity" refers to and its relevance to this study.

·       Lines 190-191: Future studies should use more robust designs, standardized metrics, and focus on caries detection and classification metrics.

·       Line 195: The discussion references the visualization of different caries characteristics in one lesion (Figure 1). Please explain what other caries characteristics are being referred to.

Comments on the Quality of English Language

Needs significant improving 

Reviewer 2 Report

Comments and Suggestions for Authors

Artificial intelligence (AI), in the near future, will be part of dental diagnosis. In this context, studies aimed at testing the validity of this technology are welcome. The objective of the study is to perform an external validation of a previously developed AI model. The research appears well conducted; however, some aspects of the paper deserve further clarification.

Introduction. A definition of early childhood caries in the introduction may help new readers better understand the research.

Method. In the selection of photographs, it is mentioned that the first step was performed by an “experienced examiner.” Information on the criteria used by the examiner would improve the clarity of the selection process.

How many dentists were in the “dental study group,” and how did they come to an agreement on the diagnosis of the evaluated photographs? The authors indicated that caries lesions were classified using the ICDAS and UniViSS criteria; however, these criteria have more categories than the ones used in the present study. Clarification of this concern is necessary. There is no information on sample size calculations. In this context, is the number of caries-free images adequate? A more detailed explanation of the origin of dental image databases is needed.

Results. Line 71 of the method section indicated that 36 images corresponded to caries-free dentitions; however, the bottom line in Table 2 indicates 37 images classified as healthy by the reference tests. Clarification is required.  

Discussion. It is written that no difference exists between AI-based caries detection results and the dental reference test. Was any statistical test performed to reach this conclusion?

According to Table 2, the sensitivity for cavitated lesions is 71.7%, a relatively low value. The clinical relevance of distinguishing between cavitated and non-cavitated caries lesions may make a comment relevant in the discussion section. The reader would benefit from an explanation of the possible impact of remineralization on the caries lesions in the IA model, considering the dynamics of the ECC process.

In light of the dynamics of the ECC process, a comment on the possible impact of remineralization of the caries lesion on the results of the IA model (sensitivity and specificity) would benefit the reader.

Round 2

Reviewer 1 Report

Comments and Suggestions for Authors

I am happy with the correction. However, I still find that the results tables are confusing and difficult to interpret. It’s unclear which data pertain to AI-based evaluation and which to visual evaluation (reference test). The cross-tabulation of AI-based image evaluation relative to visual consensus diagnosis needs to be presented more clearly.

Comments on the Quality of English Language

None

Author Response

Comments 1: I am happy with the correction. However, I still find that the results tables are confusing and difficult to interpret. It’s unclear which data pertain to AI-based evaluation and which to visual evaluation (reference test). The cross-tabulation of AI-based image evaluation relative to visual consensus diagnosis needs to be presented more clearly.

Response 1: Thank you for your comments. In order to improve clarity, we have carefully revised the result section, separated the tables and revised the corresponding text – all changes are marked-up in light blue. In addition, we have provided an additional reference that outlines the methodological background of our data handling. We hope that we have adequately addressed your concerns.